# Instagranimal: Animal Welfare and Animal Ethics Challenges of Animal-Based Tourism

**DOI:** 10.3390/ani10101830

**Published:** 2020-10-08

**Authors:** Erica von Essen, Johan Lindsjö, Charlotte Berg

**Affiliations:** 1Norwegian Institute for Nature Research Sognsveien 68, 0855 Oslo, Norway; 2Swedish Centre for Animal Welfare, SCAW and Department of Animal Environment and Health, Swedish University of Agricultural Sciences Ulls väg 26, 750 07 Uppsala, Sweden; Johan.lindsjo@slu.se; 3Department of Animal Environment and Health, Swedish University of Agricultural Sciences, P.O. Box 234, 532 23 Skara, Sweden; Lotta.berg@slu.se

**Keywords:** animal welfare, tourism, ethics, guidelines, cultural relativism, compassionate, 3Rs

## Abstract

**Simple Summary:**

Animals of countless species, wild as well as tame, can now entertain tourists on their holidays. The popularity, however, of animal-based tourism comes with significant risks for the welfare of these animals. Many animals are kept in small confinements, are broken down to interact obediently with tourists, or are made to perform, entertain, transport or even give their lives for human leisure. In this paper, the challenges of animal-based tourism are presented from the perspectives of interdisciplinary researchers. The challenges are discussed based on a two-day symposium with workshop sessions. We bring attention to the problem of cultural relativism and the difficulty of imposing universal standards of animal welfare. We conclude that reforms and individual travel decisions as a result of biosecurity concerns will impact animal welfare. In addition to this, we observe that technology has a dual role to play in enhancing edutainment but also potentially inviting new challenges. In the end, we declare some possibilities for compassionate animal based tourism.

**Abstract:**

By animal-based tourism, a host of activities offering passive viewing or active interaction with wild, semi-wild or captive animals is included. The multibillion dollar industry is on the rise globally today, offering modes of engagement with animals that trade on increasingly embodied close encounters with non-human animals. As new modes of animal-based tourism proliferate, such as sloth selfies, visiting cat cafes, swimming with sharks and agri-tourism petting zoos, animal welfare standards risk deteriorating. In the following paper, we collate concerns over animal welfare into a discussion on the challenges facing animal-based tourism. Our synthesis is the first to consider the full spectrum of such animal-based tourism: across agri-, hunting, zoo and safari tourism, to name a few, and crossing consumptive and non-consumptive boundaries. A literature review is first provided. Findings are then presented thematically following workshops at an international interdisciplinary symposium of leading tourism, animal welfare, ethics and leisure sciences scholars together with practitioners of the industry. It discusses macrolevel drivers to animal-based tourism as an industry, the problem of cultural relativism and the role of technology in enhancing or promoting the experience. We indicate ways forward toward implementing a compassionate animal-based tourism.

## 1. Introduction

Taking selfies with animals and uploading them to Instagram may be considered tacky to some but is part of a growing suite of activities in late modernity that sell embodied encounters with animals [1,2]. Today, animal-based tourism takes place in wild, semi-wild, captive or contrived settings with varying degrees of human–animal interaction [3,4]. The industry caters to tourists’ diverse taxonomical palettes: From wildlife safaris, petting zoos, having coffee and cuddles at cat cafes or volunteering with goats on farms, to “danger tourism” involving close encounters with fearsome predators [5,6], there is something for every tourist demographic and their willingness to pay, travel and get their boots dirty. 

While diverse, animal tourism activities are usually framed as authentic, tactile, multisensory encounters with natures and selves from which we have been alienated in modernity [7,8]. This explains some of the demographic characteristics of animal tourists; urban clients in particular are willing to pay a premium to “get away” from the city and rediscover the wild [1,9]. Critical perspectives have observed that when animals are conscripted into the service of tourism to provide entertainment, edutainment and self-fulfillment to tourist-consumers, they become laborers in a global capitalist economy [10]. The tourism industry often places animals in positions of risk and vulnerability, as also happens to human service industry workers [11]. Unlike the latter, however, the recent literature problematising animal predicament is new and lacking [12]. Fennell [13] suggests that the UNWTO (World Tourism Organisation) code of ethics in tourism is “decidedly anthropocentric” (p.991). Moreover, unlike human workers, animals kept in captivity do not enter into anything resembling contracts of pay and fair terms of condition. They also cannot leave these labor commitments unless, of course, they protest by attacking handlers and clients [14,15]. 

Today, few people would disagree that animals are sentient beings [16]. In recent years, more attention has been paid to the animal welfare implications of the animal-based tourism industry, and the animal welfare risks related to such activities. Owing to the work by NGOs such as World Animal Protection, research shows that tourists’ desire for close contact and high visibility of the animals they come to see typically clashes with the animals’ need for integrity [17], natural behavior, space and satisfying social environment, balanced nutrition and proper husbandry and medical care [18]. For example, Schmidt-Burbach et al. [19] found that elephants (*Elephas maximus*), pig-tailed macaques (*Macaca nemestrina*) and tigers (*Panthera tigris*) were often kept in severely inadequate welfare conditions at facilities open to tourists in Thailand. Often, animals are broken down into compliant subjects in order to interact with visitors and animals may have been removed from the wild [20] into such facilities. The welfare of domestic and semi-domestic animals is also a concern [21]. Even in instances of no ostensible physical harm, animals in contrived settings may fundamentally experience less dignity [22] and decreased welfare. Data from regulatory inspections of circus and zoo animals in Sweden, where zoo standards are nevertheless higher than in many other parts of the world, revealed non-compliance of regulatory animal welfare requirements [23]. Similarly, farmed animal sanctuaries, often lauded initiatives, may purposely breed or ensure the continued supply of orphan baby animals to attract the public [24]. Even encounters of animals in the wild, such as wildlife-spotting and swimming with dolphins (*Delphinus* sp.), may disturb individual animals and group dynamics [25,26]. 

One potential explanation for persistent poor welfare standards may simply be the limited knowledge among tourists about the normal behavior of wildlife and about how these animals are often treated in captivity [27]. Nevertheless, there are also many cultural and legislative barriers to implementing animal welfare standards today. One mechanism for continuing to uphold lax standards may be the way in which the tourism industry is set up to facilitate an ethical bleaching of one’s conduct [28]. The attitude–behavior gap in tourism [29] suggests that an individual’s positive attitude toward animal rights or environmental sustainability in their everyday life is ultimately not a reliable predictor of their holiday choices. Kline [30] argues that people leave their ethics at home while traveling because they are removed in time and place from routine and normalised contexts of everyday life. This can result in cognitive dissonance, which is managed through tropes that neutralise morally deviant behavior [31]. 

A popular idiom for neutralising cognitive dissonance is the “When in Rome” excuse, which pervades tourism culture. This means that animal welfare principles and interactions with animals become subject to cultural relativism [32]. As such, tourism comes to be seen as “a zone of permissiveness and indulgence which should not be judged by the ethical criteria deployed in daily life” [33]. Nevertheless, it also appears that some tourists are increasingly discerning consumers, whose choices and preferences on holiday signal identity. This means that changes in visitor tastes may be used to improve animal welfare standards in the future [34]. Certainly, there is appeal for many western tourists of attaching oneself to practices with epithets such as “sustainable”, “eco” and “ethical” tourism [35]. Nevertheless, there is cause also to be wary of eco-labeling and green-washing in the industry, and even words like animal “sanctuaries” [31] as these, too, function as neutralising cognitive dissonance.

### 1.1. Looking Forward

In light of such profound challenges to animal lives and welfare, what is the future of animal-based tourism? Can we expect to see a diversification in the sorts of animals commoditised, the destinations offered and array of interactions with animals available? Parallel to this and as a result of a few notable scandals (such as Cecil the Lion, Blackfish, and exposures of Phajaan, breaking of the will of elephants to get them docile in Thailand), can we also expect increased scrutiny of the industry in terms of its treatment of animal workers? A doctrine of cultural relativism prevents legislation from managing all animal-based recreational practices too uniformly, but consumers themselves may, as indicated, become more selective across practices that receive negative reviews or public naming and shaming. 

The landscape for animal-based tourism is shifting unpredictably, however, in light of COVID-19 of 2020, which saw a simultaneous drop in tolerance for practices involving interactions with wild animals and an increase in media stories on how animals in captivity were coping without visitors everyday—sometimes poorly, prompting concerns they were lonely and liked people to visit them. The lack of tourists due to the COVID-19 pandemic has had two other contradictory impacts; on the one hand, animals seem to benefit from abandoned natural areas, e.g., national parks and beaches, where they increase their presence, prompting The Atlantic to call the coronavirus “the biggest conservation action” of this time [36]. On the other hand, fewer tourists mean fewer resources and less incentive to protect wild animals and biological diversity, which may come with its own risks. In Namibia, The World Travel and Tourism Council estimates a loss of US $3.2 million in annual tourism revenue following COVID-19, an additional US $3.5 million loss of staff salaries, and increased poaching by locals who have lost their livelihoods in tourism businesses [37]. 

### 1.2. Our Symposium

Setting out to answer some of the most burning questions on the future of animal-based tourism, we arranged a two-day interdisciplinary symposium at the Swedish University of Agricultural Sciences in August 2019, titled: *Instragranimal: Animal Welfare and Ethical Challenges of Animal-Based Tourism*. The following report is a synthesis of its discussions and next steps in research, policy and practice. The symposium featured some fifty participants across disciplines and sectors, involving veterinarians, ethologists, ecologists, animal and environmental ethicists, philosophers, sociologists and tourism scholars on the one hand, and on the other hand practitioners from animal-tourism ventures and animal welfare and animal rights non-governmental organisations (NGOs). The spread of invited presenters was global, calling for researchers at the forefront of tourism studies in leading hubs in Australia and New Zealand, as well as ethologists based at the Swedish University of Agricultural Sciences. The symposium marked the gathering of these people for the first time, and also the first time that animal-based tourism was approached holistically and not divided across, e.g., prior consumptive/non-consumptive, captive/wild axes, geographical regions or segregated across industries such as hunting, agri- or ecotourism. The rationale for this was to identify common challenges across these dimensions. 

In what follows, we synthesise the discussions that were held during this symposium into four themes. The discussion was open to the public on the first day. On day two, it was organised in closed workshops for registered participants. These sessions sought to collate the topics and reflections of the two days into action points or themes for future research. The themes were chosen together with participants in plenary. They build on the ideas presented in the above introduction, discussing new arenas for animal tourism, societal drivers behind the phenomenon and compassionate animal tourism. At the end of each of these four themes, we present a short section containing next steps on three levels: directives to legislation and policy, guidelines to tourists and calls for further research. The themes reflect the main aims of the symposium, which were to look toward the future as to:Identify challenges in animal welfare and animal ethics in tourism;Identify new animal tourism developments conceptually or empirically;Explore and suggest needed regulative responses on the part of governments, international bodies or pressure from animal protection and rights NGOs and consumers of tourism, to secure the development or enforcement of welfare standards;To develop calls for future research on animal-based tourism, both intradisciplinary and across disciplines.

### 1.3. Selection of Participants

The selection of invited presenters at the symposium started with Erica von Essen and Johan Lindsjö conducting a literature review and survey of tourism research centers globally that showcased prominent researchers writing about animal-based tourism, animal welfare and ethics. A list of fifteen researchers was generated at first stage and presented as part of the application for the grant supporting the symposium. These researchers were then reached out to and personally invited by email. A desire was to span the three contexts of animal tourism: ecotourism, hunting tourism and agritourism. Media were also present during the two days and followed up on the findings afterwards, in radio, TV and a hunting journal. 

### 1.4. Synthesis of Discussion

In the following section, we present four themes of animal-based tourism that merit more discussion, research and/or legislation. These are: 1.4.1. The impact of broader societal structures on animal-based tourism, 1.4.2. Cultural Relativism—how to implement animal welfare standards globally? 1.4.3. The role of digital technology in animal-based tourism and 1.4.4. Compassionate animal-based tourism: is it possible to reconcile animal welfare with tourism?

#### 1.4.1. The Impact of Broader Societal Structures on Animal-Based Tourism

One working group at the symposium outlined the societal drivers and global processes that promote animal-based tourism on the one hand, and on the other hand the types of drivers and processes that may mediate people’s tolerance to animal suffering in these settings. Here, world events, paradigms and value shifts were discussed as to their impact on the industry going forward. A key premise in this theme may be said to be the alleviation of responsibility from the individual tourist or tourism operator, to consider macrolevel drivers responsible for the industry’s appearance today. 

What will be the long-term impact of the so-called flight shame movement on animal-based tourism abroad? It has been speculated that a decline in long-distance travel, exacerbated also by COVID-19 in 2020, may give rise to new local forms of tourism with animals. On this argument, the proximate and the everyday in the animal context may be exoticised and commoditised in, for example, staycations and day trips. This may partly account for the popularity of local agri-tourism, where nearby farms are visited [38]. These sites offer reconnecting also with local economies and pastoral culture—something that no doubt would have seemed absurd only a few decades ago. In Sweden, “cow releases” are the new agri-tourism happenings that bring families and urban residents to see local farms put their cows out to pasture for the summer grazing period, freeing them from the confinement of the winter barn The pastiche of rural life is at once two forms of “liberation”: a physical one for the cows to be enjoyed as spectacle, in terms of relieving the cows from indoor confinement and giving them the opportunity to realise themselves as bovines in the grassy field; and a spiritual one for urban visitors, who may experience the temporary alleviation of alienation from the modes of production of their dairy that they consume daily. 

However, the turn to local tourism need not necessarily be in the service of a rural renaissance. Some suggest that given the urban is becoming a significant haven for wildlife, intuitively following greater urbanisation [39] and its concentration of human–animal interaction [40]. Today, several animal tourism activities take place within the city: cat cafes and zoos, tours of the city, animal walks, but also less organised experiences such as spotting urban wildlife: pigeons in Venice, Rhesus Monkeys in Indian cities and wild boars in Berlin. While there has traditionally been a strong focus on experiencing pristine nature, in particular in so-called marketed wilderness tourism and last-chance tourism, which includes endangered species, the future may have to adapt to a “messier” nature that is characterised by multispecies interfaces, including a strong human presence [41]. Insofar as the city may be a location for such tourism, of course, there are biosecurity concerns regarding the conditioning of wildlife to feeding, trespassing human-inhabited areas, affecting sanitation and increasing zoonotic disease risk [42]. 

A future-oriented research agenda on animal-based tourism needs to consider the flows of people, the push and pull factors present in various parts of the world at any given time and may even need to attend to the long-term changes in tourism circulation, such as climate change opening up routes in the Arctic following the melting of ice caps. The increase of invasive species, posing a threat to pristine tourism based on native flagship species, may need to be problematised and reconsidered as a potential tourism revenue. 

We recognize that tourist routes may be shifted with climate change and geopolitical processes, but tourist preferences may shift independently of the physical, in longer term value shifts. As we enter a post-industrial society, there may be less emphasis on the accumulation of wealth (as has characterised agricultural societies) and more on experiences and how they contribute to a person’s identity. Given this, one might ask whether travel may increase, but travel-associated accumulation (such as physical souvenirs) may decrease. As tourism becomes a ritual context for showing identity-based goods, moreover, animal tourism experiences may be valuable to establishing a person’s status [43,44]. Within this, last-chance tourism, danger tourism and slum tourism in relation to interacting with animals may feasibly be on the rise among some tourists [3,45,46]. 

Although our symposium took place before the outbreak of COVID-19, this is a world event which will likely have profound impacts on human mobility more broadly, and where and how tourists choose to engage with animals on holiday. Biosecuritisation of borders will mean a changed travel landscape. The pandemic helped to shed a light on animal malpractices in parts of the world and generally opened the public’s eyes about the risks of human–wildlife interfaces and the dangers of keeping several animal species in confined or shared spaces for human consumption or leisure. On a fundamental level, the spatial concentration of animals into confined areas for tourism consumption involves heightened biosecurity risks [47]. Since many enterprises in animal tourism, including zoos, keep animals in these ways, change may be afoot. This may extend also to the practices of tourists, who may be compelled to adopt more clinical, hygienic ways of interacting with animals and animal-derived products, involving the washing of hands, wearing of gloves or masks, quarantining in time and space following certain encounters and handling animals or products of animals with greater care or sanitation (such as the meat from hunting trips). Research should begin to apprehend how the idea of increased biosecurity risk is impacting animal tourism, not merely from a legislative point of view, but how such risks add to, detract from or otherwise impart changes in clients’ experience of the tourism activities.

Recognising that part of the drivers for animal-based tourism is in self-fulfillment and escape of “inauthentic” lives [48], a discussion also needs to be held on the various extreme forms that help to realise such self-fulfillment. Within this, gender and masculinities play a role in enacting animal tourisms. Indeed, animal tourism may be an arena in which ideas of gender can be played out and negotiated, given that touristic settings are a liminal space partly freed from everyday constraints [49]. Moreover, that animal interactions can inform one’s gender identity seems apparent with an entertainment industry that readily commodifies macho, primeval and atavistic encounters with wild animals, where nature is marketed as a kind of antidote to the feminising influence of modern city life [50]. 

The popularity of survival shows featuring wilderness rangers, Bear Grylls and survival and self-sufficiency guides appears to testify to a masculine domain of taming the wild. Hunting packages such as the manly-titled *Scottish*, and *Alaska: Rampage*, invite a particular clientele (see, for example, the macho “shoot-outs”. Often, the animals involved in these trips are dangerous—predators or game that fight back [51]. Oppositely, contexts involving care and nursing relations with animals on holiday, including bottle-feeding baby animals or volunteering at shelters, may be both a female domain and a context in which alternative notions of masculinity can be played out. Bertella [43] suggests that direct experience of wild animals may contribute to emotional and cognitive capacities that support caring attitudes, but there is a possibility these engagements may also be pastiches—exaggerated or contrived. 

Below we present a simplified Table 1 that addresses next steps for this theme.

#### 1.4.2. Cultural Relativism—How to Implement Animal Welfare Standards Globally

Local customs and universal animal welfare standards sometimes clash. In many cases, the animal tourism industry is a significant source of income livelihood for local communities. This means that when external pressures are put on destination communities to restrict or prohibit traditional animal uses, we are presented with ethical dilemmas between human and cultural sustainability and animal welfare. 

Oh and Jackson [52] write that these instances present a dilemma between two both dominant cultural scripts in late modernity: on the one hand, one pertaining to multiculturalism, individual choice and cultural rights, and on the other hand, one pertaining to animal rights. Both, in effect, carry weight today and represent directions for policy and legislation. The latter script pertaining to animal welfare is frequently accused of cultural imperialism or ethnocentrism [53], as cultural rights and freedom of choice to animal leisure and cuisines may be a point of pride for many nationals. One can also see the dilemma as between two other cultural scripts: one of cultural globalisation and commercial homogenisation [52] and one of cultural protectionism, respectively. 

In a world where we try to simultaneously adhere to these opposing scripts, increased market and legislative pressure is placed upon parts of the world such as China where animal tourism practices are generally more permissive [54]. However, accommodating changes toward greater welfare is sometimes seen by critics as caving to international pressures [52,55]. Oppositely, scholars observe how the presumption ”in favour of right to culture” [53] that endorses the multiculturalism script now presents “…a real danger of exoticising and marginalising immigrant minorities, placing them outside of the circle of moral dialogue, criticism and community” (p. 244). A related human rights concern that may challenge animal welfare is the increased catering to persons with disabilities. For the most part, this poses no additional problems for animals, but when animal treks are made to transport obese persons across long distances, animal welfare problems may arise. Such abuse has been documented in the case of donkey rides in seaside tourism in several parts of Europe, notably Greece and Portugal [56].

The dilemma raises questions about the expediency of hard versus soft regulatory mechanisms to implement animal welfare regulations globally, relating to both governmental legislation and voluntary standards [57]. A neoliberalisation of government policy may exacerbate the establishment of regulation. In these cases, should we trust to market and, e.g., corporate social responsibility to push through improved standards of animal welfare in tourism? Where market demand is too slow in bringing about change, additional questions are raised as to who exactly will regulate, who will inspect and whom this will affect (travelers, countries, operators, etc.). Destination marketing organisations (DMOs) and NGOs may be able to operationalise locally universal codes of conduct. This “aspirational” universal code must be somewhat flexible to allow for local context. Lovelock and Lovelock [32] write that this is in line with the multifunctionality of codes of conduct, which are to educate, aspire and regulate. Moreover, this aspirational code needs to be dynamic also across time, allowing for constant evolution as conditions and priorities change. Hultsman [58] points to a needed separation between a paradigmatic (aspirational) code of ethics and an operational code of ethics. Further, without either of these codes worked out in connection with local communities with, e.g., NGOs and DMOs, such regulation is likely to do more harm than good and be seen as imperialist. 

The above presuppose a dilemma between animal welfare and human sustainability, but ways forward may include a stronger emphasis on the interrelation of human and animal welfare, as in the “One Health”, “One Welfare” concept [59]. Otherwise stated, we need to appreciate that animal welfare and human rights are not diametrically opposed. Addressing animal welfare within the Sustainable Development Goals [60] specifically may be a potential way forward in acknowledging the intersectionality of human and non-human oppressive conditions in tourism. Indeed, pitting these against one another is likely to undermine not only the welfare of both, but human–animal relations, as protected animals often become the subject of resentment among locals. Following the killing of Cecil the Lion and the uproar of the western world, some Zimbabweans expressed concern that people in the west cared more about lions than the predicament of poor Zimbabweans. 

It is also important to note that western culture does not have a monopoly on animal welfare and rights, or compassionate animal practices. Hence, attempts to implement a typically western model for other countries may be counterproductive insofar as cultures may come with their own existing and historically grounded repertoires of animal ethics, notions of stewardship and benevolence [32]. Following this, codes of conduct pertaining to animals would do well not only to list prohibitions, but to emphasise cultural virtues in dealing with animals. It may then be more effective to build on these than to implement top–down directives. It must also be recognised that there are substantial differences in moral values and animal practices also within societies [32]. As COVID-19 intimated, many Chinese may already be opposed to wildlife markets [61]. 

Below in Table 2, we present an overview of next steps for research and practice in regard to the cultural relativism challenge meeting animal-based tourism.

#### 1.4.3. The Role of Digital Technology in Animal-Based Tourism

Technology can powerfully mediate distance and interactions with animals, changing the way we see and think about animals [62]. It can bring us into heretofore unmatched proximity; new techniques such as drone-based thermal images and camera trapping invite us squarely into the everyday lives of wild animals in dens and burrows [63]. No doubt, the success of BBC’s Planet Earth series owes much to the advances in technology that enable us to learn more and get closer and into areas or animal behavior previously hidden from view. 

For this reason, it should be asked to what extent a fully or semi-virtual animal-based tourism may be on the rise today. We may also ask the extent to which this may replace or complement “real” encounters, thus taking some pressure and stress off the animals in their habitats, enclosed or wild. In times where crowding and overtourism is a real concern [64], not just for the “victims” of tourism (animals, locals, cultural heritage and property) but also as something that denigrates the tourism experience also for the clients, more remote viewing appears promising. Indeed, the use of binoculars to improve views, trail and surveillance cameras, sometimes even mounted in the nests and dens of animals, allow intimacy without getting civilians physically close to animals. The use of drones capturing footage, which can now come extremely close to many wild animals without or at least cause less disruption to their behavior, may hence allow for remote viewing close-ups. To this end, this provides only visual satisfaction for tourists, and the thrust of the embodied turn in animal-based tourism is that of advocating for a multisensory engagement that transcends mere passive viewing, involving tactile and auditory senses [6,65,66]. 

One striking development that has effected human–wildlife relations is the democratisation of the access and scope of technology used to document animal encounters. No longer the purview of professional photographers, amateurs can capture footage of wild animals with their smartphones, camera traps and take part in apps that tell them where animals are. Such technology has formed the basis for several citizen science monitoring programs [67]. Mobile phone technology can thus bring power down to the individual level, permitting decidedly personal encounters memorialised in custom photos. Taken to its extreme, the technology could also be brought down to the animal level where animal-mounted go-pro cameras show animal points of view and agency. The ability of ordinary people to autonomously zoom in on cameras affords a potentially personalised, intimate view of animal lives that is not captured in edited documentaries that often focus on grand spectacle. Moreover, future research needs to consider the use of technology from the animal side, in terms of using apps and programs to communicate their needs to us, or games on, e.g., iPads for stimulation in enclosures. Recently, for example, virtual reality goggles fitted for cows were devised to stimulate green pastures for the cows when in reality, their habitats were confined indoors. Could this technology be used to contribute to animal welfare goals in animal-based tourism or would it perhaps make it worse?

Technology also provides a divisive subject in mediating the human–animal experience in animal-based tourism. Discussing “cheater technology” [68], critics have argued that too sophisticated equipment detracts from the animal encounter. Hence, an app that lures birds to a site may be frowned upon by dedicated birders; heli-hunting is seen as “not real hunting” by hunting tourists, and the addition of technological “comforts” in nature-based trips, such as iPads and microwave ovens, may be criticised by tourists who value authenticity. Use of and acceptance of technology will vary across animal-based activities, across demographics and between different types of technology. As contended, technology used in the service of promoting the virtues and authenticity of the animal experience may be permitted, while ones that do away with these virtues may be criticised and prohibited or phased out. This coheres with scholars’ assertion that technology is neither good nor bad, instead showing an ambivalent face, being “empowering and hindering at the same time” [69]. Just as technology may alter or distort a tourism experience from some ideal type, technology may also serve to reproduce certain representations of animals as caricatures and facilitate the continued consumption of these animals in particular ways. 

Burt [70] writes that technology and visuals do not merely reflect but *constitute* animal ethics. The way we film particular species, and what we leave out for viewers, serve to shape our notions of animals. Hansen et al. [71] suggest that the public’s vocabulary for communicating about the environment is predominantly visual. Technology has thus meant greater scope for manipulating animals into televisual commodities “…packaged for the purposes of eliciting donations, membership monies, and repeat visits” which is reflected in our treatment and legislation of them. 

Social media are an intuitive context and platform for both tourism advertising and generating expectations on animal encounters and for clients potentially disseminating critical reviews and exercising moral reflection [49]. In a time of perpetual documentation of our experiences on holiday and our life achievements generally [48], animal tourism experiences are lived out again on social media and reviewed on travel and booking platforms—making animal tourism also a digitally reproduced endeavor. Here, the influence of “intermediaries” between tourists and the industries, including Expedia and TripAdvisor, play a critical role. Influencers on Instagram showing close contrived encounters with wild animals, including #slothselfies and cuddling with tiger cubs at Thai “sanctuaries”, are now recognised as an increasingly harmful driver to animal tourism, insofar as it mischaracterises animal interactions. Concepts such as “social envy”, stemming from viewing others’ holiday experiences online, and e-lineation, describing a myopic or outright harmful representation of the real thing, are now explored in relation to animal tourism [48]. Instagram has now taken action to delete images associated with certain hashtags associated with poor animal welfare, or advancing alerts for them [72]. 

Lastly, technology provides opportunities for edutainment in animal tourism: clients learning about the animals, their biology and ecology, not only through footage captured by sophisticated camera technologies, but interactively on digital platforms. They can also feature a kind of learning with animals, as through the anthropomorphising of particular animal into pedagogical figures for edutainment. Many zoos today feature interactive quizzes and ways to educate and entertain visitors. Indeed, Verma, van der Wal and Fischer [67] discuss whether technology may be part of an important bridging communication tool between policy-makers, the media, conservation practitioners and the general public. Cloke and Perkins [73] suggest cetacean encounters in dolphin tourism, both to entertain and to educate, heavily rely on an assemblage of technology: sonar, telecommunications, radar, spotter planes and more, which may be obtrusive to the dolphins. 

Here, a recurring challenge may be achieving a balance between presenting animal lives through “simulated spectacle” and the “objectivity of science” [74], balancing the public’s investment from emotion and cognition, respectively. Digital learning is not necessarily without its risks. Within the context of many action-based tourism activities such as hunting and fishing, one now no longer learns skills from family mentors to the same extent as in the past, relying instead increasingly or at least in large part on influencers and guides on social media platforms such as YouTube, for good and bad. As contended earlier, moreover, technology also allows for enhanced manipulation through editing and effects. Grazian [75] notes that animal tourism educators still have significant power in selecting the parts to be displayed and often mute key aspects of animal lives to suit the particular audience, something which they can easily do with technology. Insofar as the representation of animals and the gaze with which they are represented can powerfully constitute ethics and human–animal relationships, care needs to be taken to not misrepresent reality. Below in Table 3, an overview is provided in a table format on suggested next steps and topics for a research agenda on the role of technology in animal-based tourism. We also outline ways forward for policy and practice.

#### 1.4.4. Compassionate Animal-Based Tourism: Is It Possible to Reconcile Animal Welfare with Tourism?

A moral dilemma that meets many forms of animal-based tourism is that the lives of individual animals are essentially “sacrificed” to benefit the species. This recurs across several tourism sectors: In hunting tourism perhaps most palpably, trophy hunting has been promoted on the basis of securing biodiversity conservation [76]. It is now a prevalent slogan across hunting tourism campaigns to “be the savior” of a species by killing its individual specimens: the “kill it to save it” narrative [5], in which the revenues from hunting tourism are said to ensure the survival of endangered species. According to many hunters, therefore, the benefits outweigh the potential harms in canned hunting [77]. Although here animals individually pay the ultimate price for the survival of their wild cousins, other forms of animal tourism showcase a similar logic in which the welfare of the individual animal may suffer so that this particular individual can serve as a flagship animal for its species [78]. Thus, zoo tourism is now promoted as a necessary evil in which the lives of charismatic megafauna in particular are devoted to ambassadorships for the greater good of their species [79]. 

The thrust of this dilemma is a trade-off between individual welfare of sentient beings and species level welfare, often conceptualised as a conflict between the sentientistic and ecocentric. The difference in moral patients across these two points of departure underpins a range of conservation conflicts today. Animal welfarists accuse ecological managers of allowing the sacrifice or suffering of individual animals to save the whole. This is especially so when there is epistemic uncertainty about the consequences of our interventions on the ecosystem. Managers, on the other hand, express frustration with what they see as the short-sightedness of the welfare perspective [80]. The ecocentric, holistic perspective runs into objections on the difficulty of speaking on behalf of the good of a whole species or ecosystem, in which seemingly grandiose claims can be made that sanction the killing of individual members [81]. Nonetheless, the sentiensistic perspective comes with obvious limitations in prescribing no particular moral obligation to endangered species or the preservation of wild habitats for the sake of species conservation alone (only insofar as its members are happy): A cow has the same right to life, and living a good life, as a white rhino. Moreover, if one white rhino were to be used for spreading knowledge and awareness about the plight of his species, by being placed in a life of captivity, this cannot be condoned on the sentientistic deontological rationale. 

Given the ostensible incompatibility of these two points of departure, which one ought we to prioritise when there is a conflict? In animal-based tourism, which level of being should be ascribed moral priority? Frustrated with the trade-off, a growing body of scholars have embraced so-called compassionate conservation, which promotes the consideration of animal welfare in conservation, benefitting both individuals, species and conservation outcomes Compassionate conservation [82] is debated, mainly because of the inherent conflict between the cost for (welfare of) the individual animal and the greater good for a population or species (i.e., is it possible or not to apply compassion in successful conservation activities). Burns [83] extracted valuable lessons from an approach that embodies compassionate conservation in wildlife tourism. We further extend the concept of compassionate conservation to compassionate animal-based tourism, i.e., the welfare of the individual animal within all kinds of tourism; eco-, hunting and agritourism. Hence, we argue that individual animal welfare is compatible with responsible tourism. To this end, two approaches may be used to practice compassionate animal-based tourism: 

First, the 3Rs (*Replacement* of animals with alternative methods, *Reduction* of the number of used animals and *Refinement* of the methods, including housing and care, to mitigate suffering and promote animal welfare), initially developed to improve animal welfare for animals used in research [84], can be applied in other areas as well, including animal-based tourism. The initial question is: Is there a need to use animals at all? Is the interest for the tourism (and thus society) bigger than the cost of the individual animal? From a compassionate perspective, are there any activities where the use of animals can never be accepted? Animals can be *replaced* with virtual reality or completely replaced by tourist activities without any animal theme. If animals are involved, how many animals need to be involved in a given activity? In line with *reduction,* zoos, amusement parks, elephant and horse riding camps and farms may exhibit fewer species or fewer individual animals within a species, fewer species or individuals have to be affected by safari or trophy hunting activities, catch and release fishing, etc. *Refining* the treatment of animals used in tourism will ensure a good welfare and compassion for these animals. Housing (captive animals), exposure, care and handling that enable natural behavior, health and positive feelings will not only benefit the individual animal, but also groups, populations and species, especially if these are small or otherwise vulnerable. Compassionate animal-based tourism relies on animal protection, i.e., what we do, or ought to do, to provide a good animal welfare through legislation, but also education, policy making and, importantly, information that reaches out to tourists, industry and decision makers. 

The second approach emphasises the need of information to the tourists. Certification and labeling of products and services, including the introduction of codes of practice, are well-known strategies to inform consumers. We believe that this is one strategy to help tourists to make animal-welfare-friendly and compassionate choices when traveling. To achieve credibility in society, this needs to be based on scientific knowledge and collaboration between NGOs, academia, industry and local people involved in, or otherwise affected by, tourism. Another way to reach out to tourists is to provide information about animal-based tourism at travel hubs, i.e., airports, ferry terminals, train and bus stations, car rentals, etc. To ensure effective communication, PR strategists and experts in advertising should be involved. The benefits for animals, humans and the environment from a responsible tourism need to be communicated. The final Table 4 summarises next steps for a compassionate-based animal tourism.

## 2. Conclusions

We began by highlighting current debates in the field of animal-based tourism as this industry is gaining in popularity and scope. Animal welfare challenges were identified and traced to cognitive dissonance among tourists and to structural barriers of the industry. We noted that animals become laborers in a global capitalist economy when they are conscripted into the service of the tourism industry. The form of labor that they provide varies. In the above sections, animals featured in various roles, including:A kind of spiritual commodity [73], including serving as totem animals;A mode of transportation, such as donkey treks or dog-sledding in the Arctic;A laborer in the background to a more center-stage tourism activity [32], such as how research has found that having wolves in an area where hunting tourists hunt herbivores adds to the overall experience [45], or goats in the background to a farmers’ market;A culinary delight [30], such as alligator meat in Louisiana [5];A front-stage performer, such as animals doing tricks in circuses [27];A marker of place, such as the kangaroo or koala for Australia [11];A “facilitator” of leisure [13], such as an animal trained to serve drinks;The face of a souvenir or toy [5,43];The ultimate sacrifice as a game animal to be hunted as part of the Big Five.

Our symposium, devoted to identifying the main challenges for this industry from an ethical and welfare perspective, generated four overlapping themes worthy of further exploration: macro processes as drivers to animal-based tourism, cultural relativism as a potential challenge to implementing universal animal welfare standards, the role of technology in enhancing, promoting or even replacing animal-based tourism, and the potential for a compassionate animal-based tourism that could reconcile animals’ well-being with tourists’ interest. These are themes that point, above all, to a complex and changing landscape for animal-based leisure, with an uncertain future. A shift in public perception toward either greater reverence for animal welfare and condemnation of unethical practices, or increased concern toward host destinations and locals needing to make a living in an uncertain future, will make or break some animal-based tourisms. Likewise, based on present trends and values, we suggested that global travel patterns and pandemics will significantly impact the future of certain tourism activities. 

In addition to the above cross-cutting themes, we noted also tension between needing to balance the educational and entertainment functions of animal-based tourism, respectively. “Edutainment” seeks to provide both, but there are inherent challenges to finding an ethical, workable and profitable balance. In this regard, technology may aid in bringing us closer into animal lives and facts without becoming unduly obtrusive. This is in line with the 3Rs (*Replacement*, *Reduction* and *Refinement*). However, we also noted some risks related to technology with regard to tourists’ experience and insights. For example, online reproduction of animal-based tourism and its tendency to convey partly selective or false impressions may hence build expectations and contribute to a culture of commodification. 

Furthermore, in the suggested next steps for each of these themes or challenges for animal-based tourism, there are two levels to contend with: the structural level and that of individual clients. As evidenced, both literature and policy struggle to determine which of these levels is the most expedient to try to impart changes on. A third and final tension identified by the scholars of our symposium was that of reconciling sentientistic welfare perspectives—valuing the well-being of the individual animal—and ecocentric perspectives. Indeed, this cut across multiple tourism sectors. There is as of yet no consensus on whether it is morally right to “sacrifice” animal lives to benefit their wild species kin. Ultimately, this raises a larger discussion on the value of the wild, the loss of value of ex situ confined animals in zoos or parks.

In the end, the future of animal-based tourism may be somewhat uncertain. One might anticipate that its growing popularity may paradoxically undermine its success, as it is often predicated on selling the rare and exotic. Hence, if these products become too commonplace, animal engagement may lose part of its appeal in the future. Nevertheless, as citizens, we have a moral obligation to travel with responsibility and use compassion for animals involved in tourism. 

## Figures and Tables

**Table 1 animals-10-01830-t001:** Summary of workshop conclusions on legislation and policy, guidelines to tourists and calls for further research on the impact of broader societal structures on animal-based tourism.

**Legislation and Policy:**	Develop, review and ensure implementation of animal welfare legislation and “best practice”—guidelines in animal-based tourism, nationally and internationally.
**Guidelines to Tourists:**	Go local and explore animal-friendly and ethically justifiable animal tourism venues at home before flying across the world. Be a responsible tourist—inform yourself, contact travel retailers and tour operators, demand animal-friendly and ethically justifiable approaches to animals in tourism.
**Calls for Further Research:**	Society′s view of animals’ roles in animal-based tourism—How do the perceptions, values and attitudes of tourists correspond to those of tourism operators and animal welfare organizations? Possibilities to stimulate local, animal-welfare-friendly and ethically justifiable animal-based tourism. How gender is performed, contested and negotiated in animal encounters in animal-based tourism.

**Table 2 animals-10-01830-t002:** Summary of workshop conclusions on legislation and policy, guidelines to tourists and calls for further research on cultural relativism.

**Legislation and Policy:**	Develop, review and ensure implementation of animal welfare legislation and “best practice” guidelines in animal-based tourism among travel retailers, tour operators and animal users, emphasising the benefits from a sustainability and human perspective as well. Develop, review and implement legislation and guidelines about information, certification and labeling, implementation of codes of conduct, also including benefits from a sustainability and human standpoint.
**Guidelines to Tourists:**	Be a responsible tourist—inform yourself, contact travel retailers and tour operators, demand animal-friendly and ethically justifiable approaches (compassion—do no harm) to animals, humans and the environment in tourism (One Welfare). Push for certification, labeling and information before and during traveling, a code of conduct, based on One Welfare.
**Calls for Further Research:**	Attitudes and compliance of certification, labeling and information about animal-based tourism—potential cultural differences. The roles and responsibilities of humans in animal-based tourism. Impact on UN Sustainability goals from animal-based tourism—and its interconnection. Animal-based tourism from a One Welfare perspective.

**Table 3 animals-10-01830-t003:** Summary of workshop conclusions on legislation and policy, guidelines to tourists and calls for further research on the role of digital technology in animal-based tourism.

**Legislation and Policy:**	Outreach and education about animal welfare and ethical challenges, resulting in guidelines for web-based platforms and influencers. Certification and labeling on internet-based platforms (websites, social media) informing about and promoting animal-based tourism activities. Implement codes of conduct. Promote development and implementation of virtual animal-based tourism (see 3Rs in Section 1.4.4). Develop and implement legislation/guidelines about using animals first when technology cannot replace use of real animals.
**Guidelines to Tourists:**	Require that web-based platforms and influencers consider the animals’ situation and ethics surrounding animal use, demand that they take a standpoint (a condition for your attention, you following them, etc.). Require tour operators to consider and implement technical development replacing, reducing and refining animal use.
**Calls for Further Research:**	The impact of web-based platforms, including influencers, on animal-based tourism and how they can promote animal-friendly and ethically justifiable tourism.

**Table 4 animals-10-01830-t004:** Summary of workshop conclusions on legislation and policy, guidelines to tourists and calls for further research on compassionate animal-based tourism.

**Legislation and Policy:**	Develop, review and ensure implementation of animal welfare legislation and “best practice” guidelines (based on research on animal health, physiology, behavior, emotions and natural living) among travel retailers, tour operators and animal users. Ban non-acceptable animal activities in tourism. Develop, review and implement legislation and guidelines about information, certification, labeling and introduction of codes of conduct.
**Guidelines to Tourists:**	Be a responsible tourist—inform yourself, contact travel retailers and tour operators, demand animal-friendly and ethically justifiable approaches (compassion—do no harm) to animals in tourism. Push for certification, labeling and information before and during traveling. Require tour operators to include a 3R approach, replacing, reducing and refining animal use.
**Calls for Further Research:**	Attitudes and compliance of certification, labeling of and information about animal-based tourism—potential differences between activities, species and demography. Health, physiology, behavior, emotions and natural living with regard to different species and if and how they are suitable in animal-based tourism. Knowledge, attitudes, identification and implementation of compassion and the 3Rs in animal-based industry.

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
