# Peer review of "Instagranimal: Animal Welfare and Animal Ethics Challenges of Animal-Based Tourism"

_animals, 2020, doi:10.3390/ani10101830_

Round 1

Reviewer 1 Report

I like the revisions as whole, much more readable and organized.

There are typos in lines 102, and 503. The font of the works cited does not match the body of the manuscript

Author Response

Thank you!

We fixed both of these, as well as the erroneous capitalization on the individual words at line 503.

The font has been standardized to the same as the rest of the manuscript.

Reviewer 2 Report

The paper has been revised to take account of many of the issues raised by the reviewers. The paper is acceptable for publication.

Author Response

Thank you!

Reviewer 3 Report

This is much improved.  However line 21-23 needs some tweaking as it doesn't quite make sense: 

"We also discuss broader societal trends that impact animal-based tourism, declaring changes in mobility as a result of biosecurity concerns and the effects of technology put in the service of
enhanced edu-tainment of tourists but also impacting animal welfare."

The formatting of references needs checking as it is inconsistent (eg line 101 and 107) 

Line 155 :  "(the likely cause of the outbreak)".  This statement definitely requires an evidenced based reference - or delete it.  There are other speculative theories too....

Author Response

This is much improved.  However line 21-23 needs some tweaking as it doesn't quite make sense: 

"We also discuss broader societal trends that impact animal-based tourism, declaring changes in mobility as a result of biosecurity concerns and the effects of technology put in the service of enhanced edu-tainment of tourists but also impacting animal welfare."

We rewrote this sentence. Thank you for pointing it out. 

The formatting of references needs checking as it is inconsistent (eg line 101 and 107) 

We fixed this.

Line 155 :  "(the likely cause of the outbreak)".  This statement definitely requires an evidenced based reference - or delete it.  There are other speculative theories too....

We removed this parenthesis as it was not necessary to the point we were making. 

This manuscript is a resubmission of an earlier submission. The following is a list of the peer review reports and author responses from that submission.

Round 1

Reviewer 1 Report

A very interesting paper which hopefully will stimulate discussion and development of policy.  There could perhaps be more emphasis on the use of tourism as an opportunity to educate and widen knowledge within society, although line 116-118 acknowledges the limited knowledge and ignorance that currently exists and line 446 acknowledges the use of edu-tainment there could be more on this.  Also acknowledge that editing technology can very easily misrepresent welfare issues - either for better or for worse and this needs to be mis-used. 

There could be some more detail on the biosecurity implications of tourism activities.

Three references cited are missing from the list: Winders 2017 and Schmidt-Burbach et al 2015 (line 96), National Geographic 2017 (line 444) 

There are a number of minor typos that need correcting.

Author Response

A very interesting paper which hopefully will stimulate discussion and development of policy.  There could perhaps be more emphasis on the use of tourism as an opportunity to educate and widen knowledge within society, although line 116-118 acknowledges the limited knowledge and ignorance that currently exists and line 446 acknowledges the use of edu-tainment there could be more on this.  Also acknowledge that editing technology can very easily misrepresent welfare issues - either for better or for worse and this needs to be mis-used. 

We have added some more caution on the topic of technology being put in the service of edu-tainment. Specifically, we refer more clearly to its power to distort, omit and manipulate through editing and effects in this context. Additional references have been added about technology simulating pedagogical animal figures for edu-tainment, for children and adults, in this section, as well as an example from cetacean tourism. We have also inserted edu-tainment in the introduction now, to flag its importance.

There could be some more detail on the biosecurity implications of tourism activities.

We agree about the timeliness of this statement and have added a paragraph in the section where we before only briefly mentioned biosecurity, i.e. “The impact of broader societal structures on animal-based tourism”. Here we identify the increased biosecurity risks of confinement and concentrations of animals for tourism consumption, and point to changed practices on the part of both the industry and the individual consumers when engaging in these activities. We explore what risk might mean for the activities, insofar as it may add to, detract from, or change the experience of interacting with animals.

Three references cited are missing from the list: Winders 2017 and Schmidt-Burbach et al 2015 (line 96), National Geographic 2017 (line 444) 

We have now added these in the reference list.

There are a number of minor typos that need correcting.

The manuscript has now been proofed again. In addition to small spelling mistakes, we have fixed some sentence structure issues and corrected all English to British English.

Reviewer 2 Report

Dear authors,

overall a very interesting article, which takes up a current issue. There is merit to this paper and you have investigated a useful area of animal welfare. I carefully read your manuscript and I found the topic interesting. The paper is acceptable for publication with the suggested revision listed below.

Title: "Instagranimal” is an interesting word but in the title I think it’s unclear and not so intuitive.  

Simple summary: it is worth adding your findings

Keywords: please add “guidelines”

Tables: please add the titles

Line 72: please delete p. 984

Line 80: please delete p. 991

Line 133: please delete p. 6

Line 101 and 128 and 139: Moorhouse et al.

Line 114: in the title “Tourists behaving badly” I’d mention to the cognitive dissonance

Line 158: “.. less tourists means less resources and less incentive to protect wild animals..” please add an example

Line 321: please delete p 244

Line 345: Please delete Appleby

Line 364: “As Covid-19 made clear, a majority of Chinese are opposed to wet markets” Provide a reference.

Line 401: “VR goggles” please add details

Line 433: please delete p 110

Line 572-576: I suggest clarity here

Author Response

Dear authors,

overall a very interesting article, which takes up a current issue. There is merit to this paper and you have investigated a useful area of animal welfare. I carefully read your manuscript and I found the topic interesting. The paper is acceptable for publication with the suggested revision listed below.

Thank you very much for your helpful and nice comments.

Title: "Instagranimal” is an interesting word but in the title I think it’s unclear and not so intuitive.  

We would very much like to keep this catchy title, which also provides a link to our sympsium so people can find it. So, we now write in clearer examples to animal selfies on Instagram in the beginning of the introduction to make it explicit what we’re referring to, so that it’s not so much of a stretch any more.

Simple summary: it is worth adding your findings

We have now added a sentence on our conclusions, including the way we see that technology and biosecurity concerns are going to impact the industry.

Keywords: please add “guidelines”

Added, along with another keyword. We removed hunting.

Tables: please add the titles

Titles and short descriptions have been added to all tables. 

Line 72: please delete p. 984

Removed.

Line 80: please delete p. 991

We kept this, but made clear it was a quote, thereby adding proper quotation marks.

Line 133: please delete p. 6

This is a verbatim quote, so we keep the page number.

Line 101 and 128 and 139: Moorhouse et al.

All references have been properly updated to the journal’s referencing style.

Line 114: in the title “Tourists behaving badly” I’d mention to the cognitive dissonance

We already discuss cognitive dissonance in this section (in the same paragraph) so we’re unsure whether the reviewer wants us to work it into the title. We initially did so, but it read a bit inelegant, and we decided to remove the headline altogether.

Line 158: “.. less tourists means less resources and less incentive to protect wild animals..” please add an example

We have added an example reported by the UN from Namibia.

Line 321: please delete p 244

Again this is a verbatim quote so we keep the page number.

Line 345: Please delete Appleby

As per above, all references have been corrected to MDPI ref style.

Line 364: “As Covid-19 made clear, a majority of Chinese are opposed to wet markets” Provide a reference.

We have reformulated this as it was not technically correct. Wet markets are not the same as wildlife markets. It is the latter that many Chinese oppose. We have added a reference.

Line 401: “VR goggles” please add details

We expanded on this sentence.

Line 433: please delete p 110

Deleted.

Line 572-576: I suggest clarity here

We admit this was unhelpfully formulated and have rewritten these sentences for greater clarity.

Reviewer 3 Report

This manuscript reads as if it is actually two distinct articles. The first (p.1-p.5) being an op-ed piece and the second a summary of a symposium. The summary is cogent and should remain with minor revisions, but the first section needs major revision or should be simply eliminated and replaced with an introduction. The introductory sentence is clumsy and could be made easier to read whilst still maintaining the point that is being made. I do not understand the last sentence in the “Animal Welfare Risks” section or its connection to the rest of the paragraph. The section beginning on line 115 is written very causally and the concept needs to be developed in a more cogent manner. The statement that tourists do not know the possibility those animals in tourism could live in compromised welfare conditions, should have some evidence. The sweeping generalizations about animal welfare in the above section, still does not adequately address that “all” animal tourism facilities are bad for animal welfare. Again, I believe that pages 1 through 5 should be replaced with an introduction or simply synthesized into the introduction that begins on page 5. The style and language is vastly different in the first five pages and makes the manuscript confusing and less focused. Line 323 alludes to the transport of Tourists on the backs of Donkeys in Greece (I assume), and should be cited as it has been documented extensively. It appears there are multiple misspellings in the quote on line 425 All of the tables need to be at labeled and introduced as to what they are (i.e. Summary table of workshop I conclusions, etc...) Overall, the second section (beginning on page 5) is well written and quite comprehensive.

Author Response

This manuscript reads as if it is actually two distinct articles. The first (p.1-p.5) being an op-ed piece and the second a summary of a symposium. The summary is cogent and should remain with minor revisions, but the first section needs major revision or should be simply eliminated and replaced with an introduction. The introductory sentence is clumsy and could be made easier to read whilst still maintaining the point that is being made.

We have made a series of smaller targeted changes to this section, including some restructuring, removal of headlines, and have shortened it, so that it works more as a set-up to the symposium-informed part 2 than its own section. The first sentence is changed/removed. We tried experimenting by first placing this section after introducing the symposium, but it did not read as elegantly without it providing the necessary intro to why we held the symposium.

For that reason, most of it is in the same place, though some chunks have been moved to later sections in the paper (like animal labour, which is now in the conclusion). We hope that the reduced length, subtle shift in tone, and general improvement on the sentence structure of this section have helped the manuscript.

 I do not understand the last sentence in the “Animal Welfare Risks” section or its connection to the rest of the paragraph.

We have reformulated the sentence now and connected it better to the cases described in the same paragraph.

The section beginning on line 115 is written very causally and the concept needs to be developed in a more cogent manner.

We have removed this sentence/question entirely and got straight to the point instead.  

The statement that tourists do not know the possibility those animals in tourism could live in compromised welfare conditions, should have some evidence.

We have rephrased this somewhat to soften its generalizing tone and we also added a reference to back up the claim (Markwell 2015).

The sweeping generalizations about animal welfare in the above section, still does not adequately address that “all” animal tourism facilities are bad for animal welfare.

We are unsure what the reviewer means by this exactly. We do not believe that all animal tourism is necessarily bad for wildlife, as such is quite an extreme abolitionist position to take. We try to provide nuances examples from all sorts of practices.

Again, I believe that pages 1 through 5 should be replaced with an introduction or simply synthesized into the introduction that begins on page 5. The style and language is vastly different in the first five pages and makes the manuscript confusing and less focused.

As per our first comment, we have made various ameliorations to this section.

Line 323 alludes to the transport of Tourists on the backs of Donkeys in Greece (I assume), and should be cited as it has been documented extensively.

We have now added this example along with a reference to donkeys in seaside tourism.

It appears there are multiple misspellings in the quote on line 425

We have corrected these, thank you for pointing them out.

All of the tables need to be at labeled and introduced as to what they are (i.e. Summary table of workshop I conclusions, etc...) Overall, the second section (beginning on page 5) is well written and quite comprehensive.

Our tables have been titled and introduced.